# Noise2Score: Tweedie's Approach to Self-Supervised Image Denoising without Clean Images

**Kwanyoung Kim**[1]      **Jong Chul Ye**[1,2,3]

[1] Department of Bio and Brain Engineering
[2]Kim Jaechul Graduate School of AI
[3]Deptartment of Mathematical Sciences
Korea Advanced Institute of Science and Technology (KAIST)
{cubeyoung, jong.ye}@kaist.ac.kr

## Abstract

Recently, there has been extensive research interest in training deep networks to denoise images without clean reference. However, the representative approaches such as Noise2Noise, Noise2Void, Stein's unbiased risk estimator (SURE), etc. seem to differ from one another and it is difficult to find the coherent mathematical structure. To address this, here we present a novel approach, called Noise2Score, which reveals a missing link in order to unite these seemingly different approaches. Specifically, we show that image denoising problems without clean images can be addressed by finding the mode of the posterior distribution and that the Tweedie's formula offers an explicit solution through the score function (i.e. the gradient of loglikelihood). Our method then uses the recent finding that the score function can be stably estimated from the noisy images using the amortized residual denoising autoencoder, the method of which is closely related to Noise2Noise or Nose2Void. Our Noise2Score approach is so universal that the same network training can be used to remove noises from images that are corrupted by any exponential family distributions and noise parameters. Using extensive experiments with Gaussian, Poisson, and Gamma noises, we show that Noise2Score significantly outperforms the state-of-the-art self-supervised denoising methods in the benchmark data set such as (C)BSD68, Set12, and Kodak, etc.

## 1 Introduction

Bayesian inference, which derives the posterior probability using a prior probability and a likelihood function for the observed data, has been an important tool in statistics. This approach has been used extensively for image denoising from early ages to the modern era of deep learning. For example, in the recent unsupervised deep learning approach for image denoising using Stein's risk estimate (SURE) [1], the unknown Bayesian risk is replaced by the SURE that can be calculated from the noisy measurement so that deep neural network training is performed by minimizing it. Unfortunately, this method is sensitive to hyper-parameters, and the neural network must be retrained if the underlying noise model varies [2]. On the other hand, there has been increased research interest in image denoisers that can be trained by minimizing variants of empirical risks that are not associated with clean data. Noise2Noise [3] was the first representative approach that does not require clean data. Unfortunately, multiple noisy versions of the same images are necessary for training. To address this, self-supervised learning approaches such as Noise2Void [4], Noise2Self [5], etc. have been developed in order to use only a single noisy image. This class of approaches, which we will call Noise2X throughout the paper, are especially important for practical applications, where noiseless clean images or multiple noisy realization of the same image are difficult or impossible to collect.

35th Conference on Neural Information Processing Systems (NeurIPS 2021).

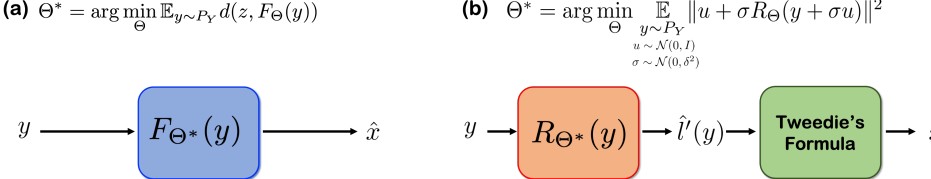

Figure 1: Overall reconstruction flow of (a) supervised learning, SURE and Noise2X, where the target $z$ and the distance measure $d(\cdot, \cdot)$ are uniquely determined by each algorithm, and (b) Noise2Score, where the first step is the estimation of the score function $\hat{l}'$ using neural network training, which is followed by Tweedie's formula to obtain the final denoising result.

Though SURE and Noise2X hold promise for practical applications, one of the fundamental questions is how these seemingly different approaches are related and whether there is a coherent mathematical theory that can be leveraged to further improve the performance. Although we are aware of a preliminary prior work [6] attempting to unite a subset of them, there is largely a lack of a principled method of designing image denoiser without clean data.

In this article, we try to approach this open problem from Bayesian statistics in a completely different route. Instead of minimizing different forms of empirical risks or SURE, one of the most important contributions of this paper is the discovery of the importance of classical result from Bayesian statistics - the Tweedie's formula [7] which provide an explicit way of computing the posterior mean of canonical parameters from the noisy measurements corrupted with exponential family noises. Specifically, we show that the Tweedie's formula provides a unified approach for image denoising from any exponential family noises through a score function (i.e. the gradient of the loglikelihood). Therefore, the self-supervised image denoising problem without clean image can be reduced to the problem of estimating the score function.

This change of perspective has many important theoretical implications as well as the flexibility with regard to the implementation of the algorithm. In fact, the score function estimation problem has been an important research topic in Bayesian statistics and machine learning [8–10]. In particular, Alain and Bengio [10] showed that the minimization of the denoising autoencoder (DAE) objective function provides an explicit way of approximating the score function. This result was further extended using the amortized residual denoising autoencoder (AR-DAE) for numerical stability and accuracy [11]. Interestingly, the training method for DAE or its variants is strikingly similar to that of Noise2X. Therefore, by combining this with the Tweedie's formula, we can obtain a novel unified framework of Bayesian approach for self-supervised image denoising. The conceptual difference and similarity of our method to the existing approaches are illustrated in Fig. 1(a)(b). As the first step of our algorithm is similar to Noise2X, we call our method Noise2Score.

Noise2Score is so powerful that it can be used to deal with any exponential family noises. Moreover, in contrast to the SURE approach, in which the network must be retrained using a different loss function depending on the noise model and parameters [1, 12], Noise2Score is universal in the sense that the identical neural network training step is used regardless of noise models. This property can be exploited to extend Noise2Score to blind setups in which noise parameters are unknown and should be estimated with minimal complexity. In addition to the novel theoretical findings mentioned above, our empirical results using additive Gaussian, Poisson and Gamma noise models have shown that Noise2Score significantly outperforms SURE and Noise2X under similar experimental conditions.

## 2 Related Works

Here we give a brief overview of the existing deep learning approaches to denoise images. As shown below, their main differences arise from the different choices of the distance metric $d(\cdot, \cdot)$ in Fig. 1.

### 2.1 Image denoising using supervised learning

Let $\mathcal{X}$ and $\mathcal{Y}$ denotes the spaces of clean and noisy images with the probability measure $P_X$ and $P_Y$, respectively. When the paired ground-truth data $x$ is available for a noisy measurement $y$, i.e.

$x := x(y)$, then the supervised training can be carried out by minimizing the following loss function:

$$\ell_{sup}(\Theta) = \mathbb{E}_{y \sim P_Y} d(x, F_\Theta(y)), \quad \text{where} \quad d(x, F_\Theta(y)) := \|x - F_\Theta(y)\|^2 \tag{1}$$

where $F_\Theta(y)$ is a deep neural network parameterized by $\Theta$ with input $y$ and here we use $l_2$ distance metric to simplify the explanation. Although this approach has significantly improved performance compared to the classical denoising algorithms [13], the supervised learning approach has a drawback as the large number of paired clean images are necessary. In real world applications, the acquisition for paired clean images are difficult or even impossible.

## 2.2 Noise2X for image denoising without clean images

To address this issue, various forms of the denoisers that can be trained without clean images have been proposed, which includes Noise2Noise [3], Noise2Void [4], Noise2Self [14], Noise2Same [5], etc. These approaches are usually based on the variants of loss functions that do not depend on clean images:

$$\ell_{Noise2X}(\Theta) = \mathbb{E}_{y \sim P_Y} d(y', F_\Theta(y)), \quad \text{where} \quad d(y', F_\Theta(y)) := \|y' - F_\Theta(y)\|^2 \tag{2}$$

where the target image $y'$ is related to $y$ in unique ways depending on algorithms. For example, in Noise2Noise [3], $y'$ is another noisy realization of the same underlying clean images. However, Noise2Noise is not practical since multiple noisy realizations of the same image are not usually available in practice. Noise2Void [14] generates $y'$ by adding blind spots on $y$. In Noise2Self [14], so-called J-invariant mask $J$ is added on the input $y$, and $y'$ is defined as the image on the complementary mask. All of these changes of the target images are intended to prevent the network from converging to a trivial identity mapping.

## 2.3 Image denoising using Stein's Unbiased Risk Estimate (SURE)

Rather than using the variants of a loss function as in Noise2X, Soltanayev et al [1] proposed a training scheme using the loss function from Stein's unbiased risk estimate (SURE) [2] which has additional regularization term. Specifically, the loss function for the SURE denoiser is defined by

$$\ell_{SURE}(\Theta) = \mathbb{E}_{y \sim P_Y} d(y, F_\Theta(y)), \quad d(y, F_\Theta(y)) := \|y - F_\Theta(y)\|^2 + 2\sigma^2 \text{div}_y F_\Theta(y) \tag{3}$$

Here, the second term in $d(y, F_\Theta(y))$ of (3) is the regularization term to prevent the network from converging to a trivial identity mapping, where $\text{div}_y$ denotes the divergence with respect to $y$. Due to the difficulty of calculating the divergence term, the authors in [1] relied on MonteCarlo SURE [15] which calculates the divergence term using MonteCarlo simulation. This introduces additional hyperparameters, on which the final results critically depend. Yet another limitation of SURE image denoiser is that the loss function is determined by the noise model. For example, in [12], the authors derived Poisson Unbiased risk estimator (PURE) to train deep CNN for dealing with Poisson noise, whose loss function is completely different from (3). Even under the same Gaussian noise models, if the noise variance $\sigma^2$ varies, the neural network $F_\Theta$ must be trained again, which causes additional computational burden. This differs from Noise2X, which uses the same loss function regardless of the noise models.

# 3 Main Contribution: Noise2Score

In contrast to the existing approaches shown in Fig. 1(a), our denoising approach is composed of two steps as illustrated in Fig. 1(b), which is inspired by the results from Bayesian statistics. In the following, we explain the details of each step.

## 3.1 Tweedie's formula of the posterior mean for image denoising

Suppose that the noisy measurement $y$ is given by

$$y = x + w, \quad \text{where} \quad w \sim \mathcal{N}(0, \sigma^2 I) \tag{4}$$

where $x$ is the clean image, and $\mathcal{N}(0, \sigma^2 I)$ denotes the independent and identically distributed (i.i.d) Gaussian distribution with zero mean and variance $\sigma^2$. Then, Tweedie's formula calculates the posterior expectation of $x$ given $y$ as [16]

$$\mathbb{E}[x|y] = y + \sigma^2 l'(y), \quad \text{where} \quad l'(y) = \nabla_y \log p(y) \tag{5}$$

where $p(y)$ denotes the marginal distribution of $y$. For the image corrupted by Gaussian noises, the posterior mean is the minimum mean square error (MMSE) estimator, so that the denoised image can be obtained using (5), as long as we know $l'(y)$ - the *score function* [17].

Tweedie's formula was extended for general class of exponential family distribution [7]. Specifically, exponential family of probability distributions are defined as those distributions whose density have the following general form:

$$p(y|\eta) = p_0(y) \exp\left(\eta^\top T(y) - \varphi(\eta)\right) \tag{6}$$

where the superscript $^\top$ denotes the transpose operation. Here $\eta$ is a canonical (vector) parameter of the family, $T(y)$ is a (vector) function of $y$, $\varphi(\eta)$ the cumulant generating function which makes $p(y|\eta)$ integrate to 1, and $p_0(y)$ the density up to a scale factor when $\eta = 0$. Exponential family distribution covers a large class of important distributions such as the Gaussian, binomial, multinomial, Poisson, gamma, and beta distributions, as well as many others. Table 1 summarizes the corresponding parameters for each exponential family distribution.

Then, using the Bayes' rule, the posterior density of $\eta$ given $y$ is given by [7]:

$$p(\eta|y) = \exp\left(\eta^\top T(y) - \lambda(y)\right)\left[p(\eta)e^{-\varphi(\eta)}\right], \quad \text{where} \quad \lambda(y) = \log\left(\frac{p(y)}{p_0(y)}\right) \tag{7}$$

where $p(y)$ and $p(\eta)$ denote the marginal distribution of $y$ and $\eta$, respectively. This implies that the posterior density is again an exponential family distribution, and the mode of the posterior distribution can be obtained by finding the maximum of $p(\eta|y)$. Specifically, by computing the gradient of $\log p(\eta|y)$ with respect to $y$ and setting it to zero, the posterior estimate of the canonical parameter $\hat{\eta}$ should satisfy the following equality:

$$\hat{\eta}^\top T'(y) \quad = \quad \lambda'(y) = -\nabla_y \log p_0(y) + \nabla_y \log p(y) = -l_0'(y) + l'(y) \tag{8}$$

where $l'(y) := \nabla_y \log p(y)$ and $l_0'(y) := \nabla_y \log p_0(y)$ are score functions, and $T'(y) = \nabla_y T(y)$. In particular, if $T(y) = y$, we can obtain the following closed form solution for the posterior mean:

$$\hat{\eta} := \mathbb{E}[\eta|y] = \lambda'(y) \quad = \quad -l_0'(y) + l'(y) \tag{9}$$

In the following, we derive denoising algorithms using Tweedie's formulation for several representative distributions.

**Additive Gaussian noise** For Gaussian distribution, in the first row of Table 1, we have

$$p_0(y) = \frac{1}{\sqrt{2\pi}\sigma}e^{-\frac{y^2}{2\sigma^2}}, \qquad \eta = \frac{x}{\sigma^2} \tag{10}$$

Accordingly,

$$\mathbb{E}[\eta|y] = \frac{\hat{x}}{\sigma^2} = \frac{y}{\sigma^2} + l'(y), \quad \text{where} \quad \hat{x} := \mathbb{E}[x|y] \tag{11}$$

since $l_0'(y) = y/\sigma^2$. Therefore, the posterior mean is given by

$$\hat{x} = y + \sigma^2 l'(y) \tag{12}$$

which is equal to (5). Even with the different parameterization of Gaussian as shown in the second row of Table 1, we can arrive at the same result using (8).

Table 1: Tweedie's formula of exponential family distribution for image denoising.

| Distribution | $p(y\|x)$ | $\eta$ | $T(y)$ | $p_0(y)$ | $l_0'(y)$ | $\hat{x}$ |
|---|---|---|---|---|---|---|
| Gaussian | $\frac{1}{\sqrt{2\pi}\sigma}e^{-\frac{(y-x)^2}{2\sigma^2}}$ | $x/\sigma^2$ | $y$ | $\frac{1}{\sqrt{2\pi}\sigma}e^{-\frac{y^2}{2\sigma^2}}$ | $-\frac{y}{\sigma^2}$ | $y + \sigma^2 l'(y)$ |
| Gaussian | $\frac{1}{\sqrt{2\pi}\sigma}e^{-\frac{(y-x)^2}{2\sigma^2}}$ | $\left[\frac{x}{\sigma^2}, -\frac{1}{2\sigma^2}\right]^\top$ | $[y, y^2]^\top$ | $\frac{1}{\sqrt{2\pi}}$ | $0$ | $y + \sigma^2 l'(y)$ |
| Poisson | $\frac{x^y e^{-x}}{y!}$ | $\log(x)$ | $y$ | $\frac{1}{y!}$ | $\simeq -\log\left(y + \frac{1}{2}\right)$ | $\left(y + \frac{1}{2}\right)\exp(l'(y))$ |
| Gamma$(\alpha, \beta)$ | $\frac{\beta^\alpha}{\Gamma(\alpha)}\left(\frac{y}{x}\right)^{\alpha-1}e^{-\beta\frac{y}{x}}$ | $\left[\alpha-1, -\frac{\beta}{x}\right]^\top$ | $[\log y, -y]^\top$ | $1$ | $0$ | $\frac{\beta y}{(\alpha-1)-yl'(y)}$ |
| Bernoulli | $x^y(1-x)^{(1-y)}$ | $\log\left(\frac{x}{1-x}\right)$ | $y$ | $1$ | $0$ | $\frac{e^{l'(y)}}{1+e^{l'(y)}}$ |
| Exponential | $xe^{-yx}, \ y \geq 0$ | $-x$ | $y$ | $1$ | $0$ | $-l'(y)$ |

**Poisson noise**   For the case of Poisson noises, we have

$$p_0(y) = \frac{1}{y!} \quad , \quad l_0'(y) = -\frac{\nabla_y \Gamma(y+1)}{\Gamma(y+1)} \simeq -\log\left(y + \frac{1}{2}\right) \tag{13}$$

where $\Gamma(y)$ denotes the gamma function and the last approximation comes from [7]. Accordingly, we have

$$\hat{\eta} = \log(\hat{x}) = \log\left(y + \frac{1}{2}\right) + l'(y) \tag{14}$$

This leads to the following posterior estimate of the image

$$\hat{x} = \left(y + \frac{1}{2}\right) \exp(l'(y)) \tag{15}$$

This corresponds to Tweedie's formula for Poisson case [16].

This result can be generalized to the low photon count sensing scenario, where the sensor measurement can be described by [18]:

$$y = \zeta z, \; z \sim \text{Poisson}(x/\zeta) \tag{16}$$

where $z$ is a random variable that follows the Poisson distribution in Table 1 and $\zeta > 0$ is the gain of the acquisition process that is related to the noise level. Then, we have

$$p_0(z) = \frac{1}{z!}, \qquad \eta = \log\left(\frac{x}{\zeta}\right) \tag{17}$$

Therefore, we have

$$\frac{\hat{x}}{\zeta} = \left(z + \frac{1}{2}\right) \exp(l'(z)) \quad \longrightarrow \quad \hat{x} = \left(y + \frac{\zeta}{2}\right) e^{l'\left(\frac{y}{\zeta}\right)} \tag{18}$$

**Gamma noise**   Gamma noise distribution can be used to model the speckle noises in various imaging application. Specifically, the image corrupted with the speckle noise is represented by

$$y = xn, \quad n \sim p(n; \alpha, \beta) = \frac{\beta^\alpha}{\Gamma(\alpha)} n^{\alpha-1} \exp(-\beta n) \tag{19}$$

where $p(n; \alpha, \beta)$ denotes the Gamma distribution with $(\alpha, \beta)$ parameters. Here, $(\alpha, \beta)$ are the parameters which determine the noise level of Gamma noise distribution. For example, for the $k$-look measurement case, $\alpha = \beta = k$ [19]. Then, the probability $p(y|x)$ can be obtained by replacing $n$ in $p(n; \alpha, \beta)$ with $n = y/x$, and the resulting density function can be bound in Table 1. Then, we can easily show that

$$p_o(y) = 1, \quad \eta = \begin{bmatrix} \alpha - 1 \\ -\beta/x \end{bmatrix}, \quad T(y) = \begin{bmatrix} \log y \\ y \end{bmatrix} \tag{20}$$

Using (8), we have

$$\frac{\alpha - 1}{y} - \frac{\beta}{x} = l'(y) \quad \Longrightarrow \quad \hat{x} = \frac{\beta y}{(\alpha - 1) - y l'(y)} \tag{21}$$

Similar derivations for other exponential family distribution can be found in Table 1.

## 3.2   Score function estimation

So far, our derivation assumes the prior knowledge of the score function. In practice, this should be estimated, so here we describe how the score function can be estimated using a neural network.

Historically, Hyvärinen et al [8] was the first to derive a remarkable implicit score matching objective that no longer requires having an explicit score target but is nevertheless equivalent to the original problem that minimizes the expected quadratic distance between the model function and the score function of data. With the advance of the Denoising Auto Encoder (DAE) [20], the author of [9] observed that the minimization of the DAE objective function with the residual form is related to the score matching between the model and perturbed data. This observation was rigorously analyzed

by Alain and Bengio [10], who showed that as the perturbed noise becomes sufficiently small, DAE leads to the score function.

More specifically, in the denoising autoencoder (DAE), the following loss is minimized:

$$\ell_{DAE}(\Theta) = \mathop{\mathbb{E}}_{\substack{y \sim P_Y \\ u \sim \mathcal{N}(0,I) \\ \sigma_a \sim \mathcal{N}(0,\delta^2)}} \|y - F_\Theta(y + \sigma_a u)\|^2 \tag{22}$$

According to [10], the optimal DAE $F_{\Theta^*}(x)$ can be represented by

$$F_{\Theta^*}(y) = y + \sigma_a^2 l'(y) + o(\sigma_a^2), \tag{23}$$

where $l'(y)$ is the score function defined in (5) and $o(\cdot)$ denotes the small "o" notation. In other words, for sufficiently small $\sigma_a$, we can approximate the score function as:

$$\hat{l}'(y) = \frac{F_{\Theta^*}(y) - y}{\sigma_a^2} \tag{24}$$

One of the downsides of the score function estimation using DAE is that the score function estimate in (24) is numerically unstable as $\sigma_a \to 0$. To address the numerical instability and reduce the approximation error, Lim et al [11] recently proposed so-called the amortized residual DAE (AR-DAE). Specifically, AR-DAE is trained by minimizing the following the objective function:

$$\ell_{AR-DAE}(\Theta) = \mathop{\mathbb{E}}_{\substack{y \sim P_Y \\ u \sim \mathcal{N}(0,I) \\ \sigma_a \sim \mathcal{N}(0,\delta^2)}} \|u + \sigma_a R_\Theta(y + \sigma_a u)\|^2 \tag{25}$$

where $R_\Theta$ is from the residual form of the DAE:

$$F_\Theta(y) = \sigma_a^2 R_\Theta(y) + y \tag{26}$$

By plugging (26) in (25), we can obtain the original DAE cost in (22) up to a scale factor. Furthermore, using (24), we have

$$\hat{l}'(y) = \frac{F_{\Theta^*}(y) - y}{\sigma_a^2} = R_\Theta(y) \tag{27}$$

Therefore, the neural network trained with the AR-DAE (25) is a direct and stable way of estimating the score function. Therefore, we employ the AR-DAE to estimate the score function from the noisy measurement as the first step of our method (see Fig. 1(b)).

## 3.3 Relation to Noise2X and SURE

Although our Noise2Score was derived from a completely different perspective, it turns out that it has very important connections to Noise2X and SURE.

Specifically, if the noisy image $y$ is corrupted by additive Gaussian, using Tweedie's fomula for Gaussian noises in Table 1, the denoised image can be approximated by

$$\hat{x} = y + \sigma^2 \hat{l}'(y) = F_{\Theta^*}(y) \tag{28}$$

where we use (27). Recall that $F_{\Theta^*}(y)$ is the neural nework trained with DAE loss function in (22). The DAE training with (22) is basically inserting noises into the images and then find the mapping that removes the noises. In terms of adding extra noise to a noisy image, our method is closely related to Noisier2Noise [21]. However, in the training phase of Noisier2Noise, the main assumption is that noise from the same noise statistic as the original noisy image should be injected into the noisy images so that the neural networks learn to reduce the noise from the noisy input image. In our Noise2Score training method, however, Gaussian noise is added to noisy images at different noise levels regardless of the noise statistics to estimate the score function rather than noise. In addition, in contrast to Noisier2Noise, our method has a post-processing step with the Tweedie's formula, which is determined by the noise model such as Poisson, Gamma, etc.

Yet another important connection is its relation to SURE. Using the residual representation in (26), the SURE cost function in (3) can be equivalently represented by

$$
\begin{aligned}
\ell_{SURE}(\Theta) &= \mathbb{E}_{y \sim P_Y} \left\{ \|y - F_\Theta(y)\|^2 + 2\sigma^2 \mathrm{div}_y F_\Theta(y) \right\} \\
&= \mathbb{E}_{y \sim P_Y} \left\{ \sigma^4 \|R_\Theta(y)\|^2 + 2\sigma^4 \mathrm{div}_y R_\Theta(y) \right\} + 2\sigma^2 \mathrm{dim}(y)
\end{aligned} \tag{29}
$$

where $\dim(y)$ is the dimension of the vector $y$, which is constant. Up to the scaling factor, this cost function is identical to the implicit score matching objective by Hyvärinen et al [8], which is given by

$$\ell_{ISM}(\Theta) = \mathbb{E}_{y \sim P_Y} \left\{ \frac{1}{2} \|\Psi_\Theta(y)\|^2 + \operatorname{div}_y \Psi_\Theta(y) \right\} \tag{30}$$

where $\Psi_\Theta(y)$ is the score function estimate parameterized by $\Theta$. Therefore, using the residual transform in (26), the SURE objective is nothing but to find the score function, which corresponds to the first step of our Noise2Score. Furthermore, due to the use of (26), SURE is not optimal for noise models other than Gaussian. Although there are variants of SURE for other noise type [22], the cost function of this method varies depending on the type of noise. However, as discussed in detail later, our method uses the same cost function for neural network training, but only the post-processing step differs depending on the type of noise. Moreover, considering that the modern research trend of the score function estimation has quickly evolved from the implicit score matching to DAE or AR-DAE due to the numerical instability and difficulty of computing the divergence term in (30), we can easily expect that the SURE denoiser also suffers from similar limitations.

### 3.4 Universal neural estimation for blind denosing problems

Recall that Noise2Score is composed of two steps: the score function estimation by minimizing the AR-DAE loss function, after which the final denoised image is obtained by applying Tweedie's formula shown in Table 1. This decoupling allows Noise2Score to have unique advantages compared to the conventional approaches.

Specifically, one of the most important advantages of Noise2Score is that AR-DAE training is universal in the sense that the same loss function is used regardless of noise models and parameters. More specifically, in (25), the model and noise parameter dependency is only through the sampling $y$ from $P_Y$. This sampling step can be performed by random patch cropping from the input images, which procedure is identical regardless of $P_Y$ being Gaussian, Poisson, Gamma, etc.

Yet another advantage is that this property can be utilized to estimate the unknown noise parameters without retraining neural networks. For example, in the case of Gaussian denoising in Table 1, we have the following form of the denoised images:

$$\hat{x}(\sigma) = y + \sigma^2 \hat{l}'(y) \tag{31}$$

If the noise parameter $\sigma$ is unknown, then we can find the optimal $\sigma^*$ value by solving the optimization problem

$$\sigma^* = \arg \min_\sigma Q\left(\hat{x}(\sigma)\right) \tag{32}$$

where $Q(\cdot)$ is an image quality penalty such as total variation (TV). In contrast to the SURE, where the $\sigma$ dependency exists in the loss function in (3) and the neural network must be trained with a new $\sigma$ value, the optimization problem in (32) is computationally negligible since the precalculated score function estimate $\hat{l}'(y)$ is used regardless of different $\sigma$ values. A similar on-the-fly parameter estimation can be used for Gamma noise model. As shown in (21), the parameters $(\alpha, \beta)$ can be adjusted using the precomputed score function estimate $\hat{l}'(y)$. Even for the Poisson noise model, the same trained neural network is used to compute the score function for each input $y/\zeta$. Therefore, our Noise2Score has advantages for the blind application where existing methods cannot be used due to the difficulty of generating samples by changing the noise level parameters without retraining.

## 4  Experimental Results

**Dataset and Implementation detail**   We evaluated the proposed method for gray-scale and color images in the four benchmark datasets: the gray-scale image dataset contains BSD68 [23] and Set12. The color image dataset contains RGB natural images CBSD68 [23], Kodak dataset. We adopt DIV2K [24], and (C)BSD400 dataset as train data set. We generated the synthetic noise images for each noise distribution. In order to evaluate the proposed method fairly with comparison methods, the same modified U-Net generator [3] is used for all methods. The total epoch was set to 100 and the Adam optimizer [25] was used to train the network. The learning rate was initialized to $2 \times 10^{-4}$ for first 50 epoch, and after 50 epoch the learning rate was decayed to $2 \times 10^{-5}$. The proposed method

Table 2: Quantitative comparison for various noise model using various methods in terms of PSNR(dB) when the noise parameters are known or unknown (N2V: Noise2Void, N2S: Noise2Self, N2N: Noise2Noise, SL: supervised learning, Anscombe: Anscombe method for BM3D).

| Noise type | | Known parameters | | | | | | | Unknown parameters | | |
|---|---|---|---|---|---|---|---|---|---|---|---|
| Gaussian | Dataset | BM3D | N2V | N2S | SURE | Ours | N2N | SL | Ours | N2N | SL |
| $\sigma = 25$ | BSD68 | 28.59 | 26.77 | 28.28 | 28.78 | **29.12** | 29.18 | 29.20 | 28.95 | 28.98 | 28.98 |
| | Set12 | 29.96 | 27.56 | 29.16 | 29.13 | **30.13** | 30.33 | 30.36 | 30.08 | 30.08 | 30.08 |
| | CBSD68 | 30.56 | 29.22 | 30.05 | 30.23 | **30.85** | 31.10 | 31.10 | 30.78 | 30.91 | 30.91 |
| | Kodak | 31.68 | 30.02 | 30.53 | 30.75 | **31.89** | 32.20 | 32.20 | 31.78 | 31.96 | 31.96 |
| $\sigma = 50$ | BSD68 | 25.62 | 24.34 | 25.61 | 25.80 | **26.21** | 26.27 | 26.30 | 25.81 | 25.86 | 25.98 |
| | Set12 | 26.33 | 24.68 | 26.19 | 26.23 | **27.16** | 27.20 | 27.20 | 26.65 | 26.65 | 26.72 |
| | CBSD68 | 27.38 | 25.13 | 27.05 | 26.24 | **27.75** | 27.94 | 27.95 | 27.32 | 27.66 | 27.68 |
| | Kodak | 27.02 | 25.75 | 28.01 | 26.93 | **28.83** | 29.07 | 29.10 | 28.13 | 28.70 | 28.71 |
| Poisson | Dataset | Anscombe | N2V | N2S | PURE | Ours | N2N | SL | Ours | N2N | SL |
| $\zeta = 0.01$ | BSD68 | 30.51 | 28.73 | 29.76 | 30.16 | **30.81** | 30.89 | 30.98 | 30.63 | 30.91 | 30.93 |
| | Set12 | 31.20 | 30.06 | 30.47 | 30.69 | **31.58** | 31.70 | 31.79 | 31.42 | 31.67 | 31.70 |
| | CBSD68 | 32.40 | 31.85 | 31.04 | 32.30 | **32.61** | 33.01 | 33.01 | 32.23 | 32.94 | 32.95 |
| | Kodak | 33.13 | 32.98 | 32.24 | 33.01 | **33.41** | 33.91 | 33.91 | 32.96 | 33.85 | 33.87 |
| $\zeta = 0.05$ | BSD68 | 26.77 | 26.12 | 26.54 | 24.76 | **27.12** | 27.23 | 27.25 | 26.77 | 27.19 | 27.20 |
| | Set12 | 27.54 | 27.21 | 27.53 | 25.07 | **27.86** | 28.02 | 28.04 | 27.76 | 28.01 | 28.03 |
| | CBSD68 | 28.33 | 28.37 | 28.32 | 26.68 | **28.68** | 29.23 | 29.27 | 28.23 | 29.07 | 29.07 |
| | Kodak | 29.31 | 29.56 | 29.52 | 26.38 | **29.71** | 30.33 | 30.40 | 28.98 | 30.13 | 30.25 |
| Gamma | Dataset | | N2V | N2S | | Ours | N2N | SL | Ours | N2N | SL |
| $k = 100$ | BSD68 | - | 29.32 | 30.49 | - | **32.67** | 32.87 | 32.93 | 32.54 | 32.83 | 32.88 |
| | Set12 | - | 30.54 | 30.71 | - | **33.01** | 33.21 | 31.72 | 32.89 | 33.19 | 33.21 |
| | CBSD68 | - | 31.11 | 30.54 | - | **33.82** | 35.45 | 35.53 | 33.50 | 35.33 | 35.33 |
| | Kodak | - | 31.96 | 31.60 | - | **34.22** | 36.26 | 36.41 | 33.82 | 36.16 | 36.16 |
| $k = 50$ | BSD68 | - | 26.98 | 29.25 | - | **30.53** | 31.10 | 31.16 | 30.42 | 30.64 | 30.64 |
| | Set12 | - | 27.36 | 29.67 | - | **30.87** | 31.68 | 31.72 | 30.83 | 31.58 | 31.67 |
| | CBSD68 | - | 30.51 | 30.19 | - | **31.05** | 33.52 | 33.53 | 30.93 | 33.40 | 33.41 |
| | Kodak | - | 31.38 | 31.03 | - | **31.34** | 34.49 | 34.57 | 31.32 | 34.39 | 34.40 |

was implemented in PyTorch [26] with NVidia GeForce GTX 1080-Ti. The network training took about 10 hours. The more detail of implementation are described in Supplementary Material. To deal with blind noise case, as for the quality metric $Q(\cdot)$ we use the TV norm for the Gaussian noises and its combination with data fidelity term for the Poisson and Gamma noise. The details of the quality metric and the discussion on the accuracy of the blind parameter estimation is provided in Supplementary Material.

**Gaussian noise** Using additive Gaussian noise, our method was compared to (C)BM3D [13], Noise2Void, Noise2Self, SURE, Noise2Noise and supervised learning as shown in Table 2. As expected, the supervised learning and Noise2Noise using multiple noisy realization were the best, but these methods are not practical as explained before. Among the other self-supervised learning approaches, our Noise2Score provide the best performance for all dataset. With the comparison with the supervised learning and Noise2Noise, the performance difference of our method is only margin. Even for blind-noise cases where the noise levels are randomly sampled from Gaussian distribution with $\sigma \in [5, 55]$, Noise2Score with the on-the-fly parameter estimation performed comparable to the supervised learning approach. These results indicate the superior performance of our method. The qualitative comparison in Fig. 2 show that Noise2Score retains the better image details and provides much visually pleasing results compared to other top two self-supervised learning approaches.

**Poisson noise** For the Poisson noise case, BM3D and SURE were replaced with Anscombe (BM3D+VST) [27] and PURE [12], respectively, as indicated in Table 2, as they are optimized for Possion noise model. Again, our Noise2Score results in significant performance gain compared to other approaches, and the results are comparable to supervised learning approach and Noise2Noise. In addition, to perform experiments when the noise parameters are unknown, we performed experiments with the noise levels randomly sampled between $\zeta \in [0.001, 0.1]$. We found that our proposed method

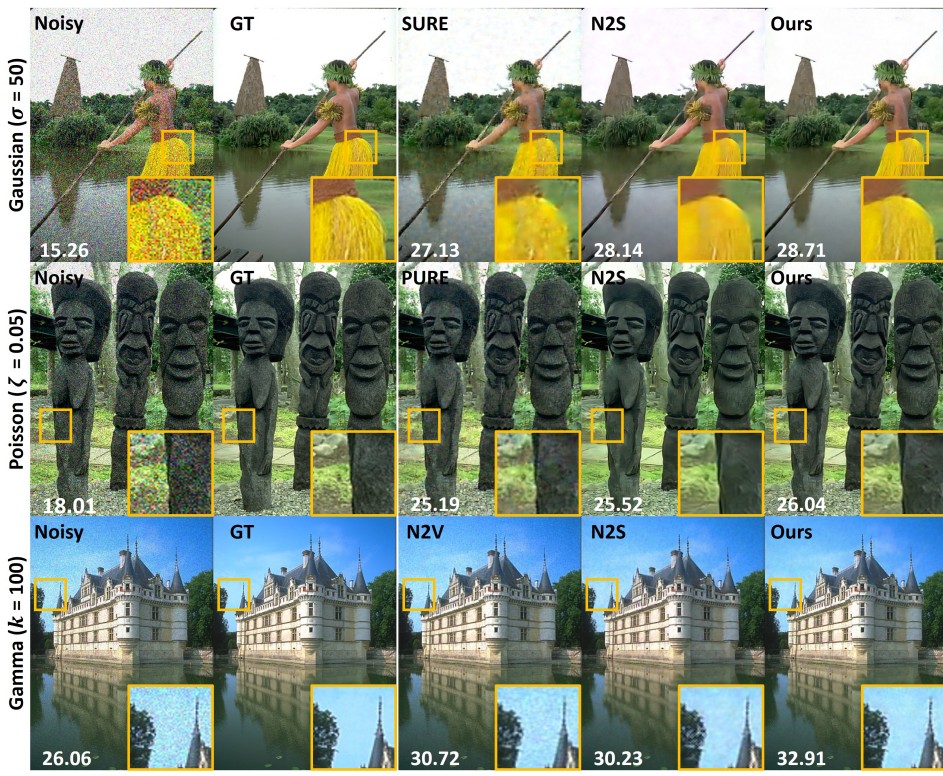

Figure 2: Qualitative comparison using CBSD68 dataset. Top : Gaussian noise with $\sigma = 50$. Middle: Poisson noise with $\zeta = 0.05$. Bottom: Gamma noise with $k = 100$. White numbers at the lower left part of the images indicate the PSNR values in dB. Noisy: noisy input, GT: ground-truth image.

produce even higher PSNR than the results of other existing methods even with known parameters, and also provide comparable results to the supervised learning approaches. The qualitative comparison in Fig. 2 with the other two best approach showed that our proposed method provides much cleaner images.

**Gamma noise**    As the extension of BM3D and SURE are not available for Gamma noises, only four comparison methods was adopted for comparison as indicated in Table 2. Here, we set $\alpha = \beta = k$. Again, Noise2Score produced best results among self-supervised learning approaches, and provided comparable results to Noise2Noise and supervised learning approaches. Even for the blind noise case, where the noise parameter $k$ randomly selected in $k \in [40, 120]$, Noise2Score provided the better PSNR results compared to the other self-supervised learning method even with known parameters, and produces comparable results to supervised learning approaches. The qualitative comparison in Fig. 2 confirm that our Noise2Score provides best reconstruction results.

## 5    Conclusion

In this work, we provided a novel Bayesian framework for self-supervised image denoising without clean data, which surpasses SURE, PURE, Noise2X, etc. Our novel innovation came from the Tweedie's formula, which provides explicit representation of denoise images through the score function. By combining with the score-function estimation using AR-DAE, our Noise2Score can be applied to image denoising problem from any exponential family noises. Furthermore, an identical neural network training can be universally used regardless of the noise models, which leads to the noise parameter estimation with minimal complexity. The links to SURE and existing Noise2X were also explained, which clearly showed why our method is a better generalization.

## Limitation and negative societal impacts

As a negative societal impact, the failure of image denoising methods could produce side effects. For example, removing both the noise and the texture of the medical images could lead to misdiagnoisis. While the proposed Noise2Score has merits, there are also limitations. In real environments, the prior knowledge of noise distribution may not be available, and the noise model could not be modeled by exponential family noises. Therefore, future extension to such scenario would be beneficial.

## Acknowledgment

This research was funded by the National Research Foundation (NRF) of Korea grant NRF-2020R1A2B5B03001980, ETRI (Electronics and Telecommunications Research Institute)'s internal funds [21YR2500, Development of Digital Biopsy Core Technology for high-precision Diagnosis and Therapy of Senile Disease], and KAIST Key Research Institute (Interdisciplinary Research Group) Project. This work was also supported by Institute of Information & Communications Technology Planning & Evaluation (IITP) grant funded by the Korea government (MSIT) (No.2019-0-00075, Artificial Intelligence Graduate School Program (KAIST)).

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
