# Supplementary Material
# Noise2Score: Tweedie's Approach to Self-Supervised Image Denoising without Clean Images

**Kwanyoung Kim**[1]       **Jong Chul Ye**[1,2,3]
[1] Department of Bio and Brain Engineering
[2] Kim Jaechul Graduate School of AI
[3] Deptartment of Mathematical Sciences
Korea Advanced Institute of Science and Technology (KAIST)
{cubeyoung, jong.ye}@kaist.ac.kr

## 1 Algorithm of Noise2Score

Algorithm 1 details the overall pipeline of the Noise2Score. First, the neural network $R_\Theta$ was trained by minimizing $\ell_{AR-DAE}(\Theta)$ to learn the estimation of the score function from the noisy input $y$. This neural network training step is universally applied regardless of noise distribution $\eta$. In particular, during the training phase, we annealed $\sigma_a$ from $\sigma_a^{max}$ to $\sigma_a^{min}$ to stably train the network as suggested in [1]. After training the network $R_\Theta$, we estimated the clean images for each noise distribution by using Tweedie's formula as reported in Table 1 in the main paper.

---

**Algorithm 1:** Noise2Score

**Given:** learning rates $\rho$, number of epochs $N$;
**Input**     : noisy input $y$ from training data set $D_\eta$ with size $m$ and noise level parameter
            $\eta \in (\sigma, \zeta, k)$, neural network $R_\Theta$, annealing sigma $\sigma_a \in [\sigma_a^{min}, \sigma_a^{max}]$;
**for** $n = 1$ *to* $N$ **do**
  $\quad u \sim \mathcal{N}(0,1)$;
  $\quad q \to n/m$;
  $\quad \sigma_a \to \sigma_a^{max} * (1-q) + \sigma_a^{min} * q$
  $\quad \ell_{AR-DAE}(\Theta) = \underset{\substack{y \sim P_Y \\ u \sim \mathcal{N}(0,I), \sigma_a \sim \mathcal{N}(0,\delta^2)}}{\mathbb{E}} \| u + \sigma_a R_\Theta(y + \sigma_a u) \|^2$;
  $\quad \Theta \leftarrow \Theta - \rho \nabla_\Theta \ell_{AR-DAE}(\Theta)$;
**end**
**Output:** estimated the score function, $R_\Theta(y) = \hat{l}'(y)$
**Inference :**
**if** Gaussian noise **then** $\hat{x} = y + \sigma^2 l'(y)$;
**else if** Poisson noise **then** $\hat{x} = \left( y + \frac{\zeta}{2} \right) e^{l'\left(\frac{y}{\zeta}\right)}$;
**else if** Gamma noise **then** $\hat{x} = \frac{\beta y}{(\alpha-1) - y l'(y)}$;

---

## 2 Quality metric for blind noise removal

To deal with blind noise removal case where the noise parameters are unknown, the image quality penalty $Q(\cdot)$ should be defined for each noise distribution as stated in the main paper. Quality metrics was built upon the total variation norm, but the data fidelity term of noise distribution was combined

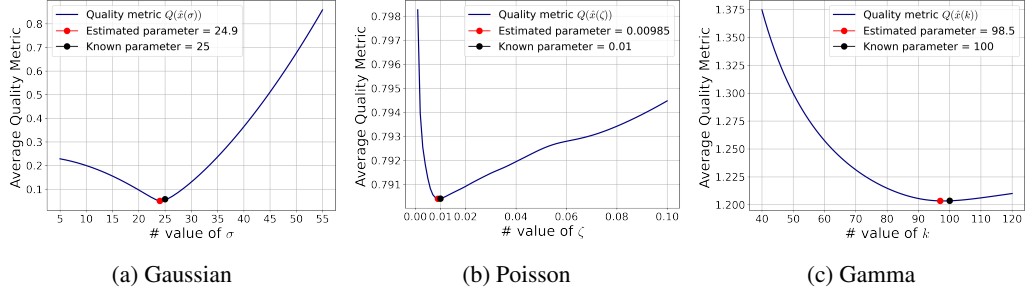

|  (a) Gaussian | (b) Poisson | (c) Gamma |

Figure 1: The average quality penalty value with respect to the noise level parameter such as $\sigma, \zeta, k$ in the Set12 dataset. (a) Gaussian noise with $\sigma = 25$, (b) Poisson noise with $\zeta = 0.01$, (c) Gamma noise with $\alpha = \beta = k = 100$. The red dot indicate the estimated noise level parameter by our method, whereas the black dots indicate the ground-truth parameters.

with total variation in the case of Poisson and Gamma noise as suggested in [2, 3]. The details are as follows.

**Additive Gaussian noise** For the addtivie Gaussian noise case, we design the quality metric by using just the total variation norm:

$$Q(\hat{x}(\sigma)) = |\nabla \hat{x}(\sigma)| \tag{1}$$

Fig. 1a shows the results of the estimation of an unknown parameter for Gaussian noise $\sigma = 25$. Esitmated parameter $\sigma^*$ from quality metric for Gaussian noise, $Q(\hat{x}(\sigma))$, is 24.8 compared to target parameter $\sigma = 25$, which confirms that our blind noise parameter estimation is quite accurate.

**Poisson noise** In case of the Poisson noise, we employ the existing total variation regularization method that jointly minimizes the negative log likelihood of the prior distribution from the EM algortihm [2]. Specifically, the quality metric for Poisson noise $Q(\hat{x}(\zeta))$ was defined as follow:

$$Q(\hat{x}(\zeta)) = \alpha |\nabla \hat{x}(\zeta)| + \hat{x}(\zeta) - y \log \hat{x}(\zeta) \tag{2}$$

where $\alpha$ is the hyperparameter for the weighting the total variation norm. Following the paper [2], the $\alpha$ value was set to 0.25, 0.1 in the case of $\zeta = 0.05, 0.01$, respectively. Fig. 1b shows the results of the estimation of an unknown parameter for Poisson noise $\zeta = 0.01$. Estimated parameter $\zeta^*$ from quality metric for Poisson noise, $Q(\hat{x}(\zeta))$, is 0.00985 compared to target parameter $\zeta = 0.01$. This again confirms the accuracy of our blind parameter estimation.

**Gamma noise** For the Gamma noise, we employ the method from the MAP estimation of Gamma noise distribution [3]. Specifically, the quality metric for Gamma nose with $\alpha = \beta = k$, $Q(\hat{x}(k))$, was defined as follow:

$$Q(\hat{x}(k)) = |\nabla \hat{x}(k)| + \alpha \frac{y}{\hat{x}(k)} + \frac{\beta}{2} \left( \frac{y}{\hat{x}(k)} \right)^2 + \gamma \log \hat{x}(k) \tag{3}$$

where $\alpha, \beta, \gamma$ are the hyperparameter for weighting each term. In [3], the authors reported that when $\alpha + \beta = \gamma$ the results are stable, so that we follow this condition and each parameter was set to 0.5, 0.5, and 1, respectively. Fig. 1c shows the results of the estimation of an unknown parameter for Gamma noise $k = 100$. Estimated parameter $k^*$ from quality metric for Gamman, $Q(\hat{x}(k))$, is 98.5 compared to target parameter $k = 100$. From Fig 1c, we found that our proposed quality metric for each noise distribution was quite good to find the optimal parameter to get the best performance in case of the blind noise.

## 3   Detail of Dataset

We adopted DIV2K and CBSD400 dataset as the trainset. We generated the cropped patch $128 \times 128$ size to train the network. For the data augmentation, we used the random horizontal, vertical filp and flop methods. To evaulate the proposed Noise2Score, the testset was adopted for Kodak, CBSD68, Set12. In case of the BSD68 dataset, we transform the CBSD68 datset into gray-scale images. The URL for each dataset is given by:

DIV2K : https://data.vision.ee.ethz.ch/cvl/DIV2K/

CBSD400 : https://www2.eecs.berkeley.edu/Research/Projects/CS/vision/bsds/

Kodak : http://www.cs.albany.edu/ xypan/research/snr/Kodak.html/

Set12 : https://www.researchgate.net/figure/12-images-from-Set12-dataset/fig11/338424598

# 4 Implementation Detail

In the training phase, we only selected one noise distribution. For example, if we train Noise2Score for Gaussian noise, all training images are corrupted by Gaussian noise. In the case of the "known" parameter, during the training we sampled noisy images with only one noise level, but in the case of experiments with unknown noise parameters, we randomly sampled images with multiple noise levels during training, as reported in the main paper. In orther wordks, the multiple trained U-Net are required to deal with the multiple cases of the combined the noise model and the noise level. For the annealing sigma $\sigma_a$ which are need to learn the score function of noisy data, we linearly decrease the perturbed noise from $\sigma_a^{max}$ to $\sigma_a^{min}$. In the cases of the Gaussian and Gamma noise, $\sigma_a^{max}$ and $\sigma_a^{min}$ are set to [0.1, 0.001], respectively. Due to the normalized issue of intensity range, we set $\sigma_a^{max}$ and $\sigma_a^{min}$ to [1, 0.05] for the Poisson noise case.

# 5 Comparison with More Baseline Algorithms

To show the effectiveness of proposed Noise2Score, we additionally carried out experimental comparisons with Laine et al [4], Noisier2Noise [5] and Noise2Same[6]. For a fair comparison, we trained all methods with the combined DIV2K and BSD400 dataset in gray scale. Then, the trained networks were used for inference using the BSD68 dataset for Gaussian noise and Poisson noise, as shown Table 3. The results confirmed that our method outperformed the existing ones including two different implementations of Laine et al, Noisier2Noise and Noise2Same.

Table 1: Quantitative comparison for various noise model using various methods in terms of PSNR(dB) when the noise parameters are known (N2V: Noise2Void, N2S: Noise2Self, Nr2N: Noisier2Noise, N2Same: Noise2Same).

| Noise type | N2V | N2S | Nr2N | N2Same | Laine19-mu | Laine19-pme | Ours |
|---|---|---|---|---|---|---|---|
| Gaussian ($\sigma = 25$) | 26.27 | 28.28 | 28.01 | 28.00 | 28.13 | 29.04 | **29.12** |
| Poisson ($\zeta = 0.01$) | 28.73 | 29.73 | - | 29.32 | 28.49 | 30.70 | **30.81** |

# 6 Application on Real Data

Although the proposed Noise2Score has difficulties in application in real data sets, we have shown the potential of Noise2Score for real application. We carried out the experiments with real fluorescence microscopy data sets (FMD) as shown Table 4. We have taken raw noisy images from the confocal FISH categories, which consisted of 1000 images. We used the 900 images as a training data set and the remaining 100 images as a test data set. In the FMD data set, the noisy images are modeled with mixed Poisson-Gaussian noise. Accordingly, we used a two-step approach in which the Gaussian noise is first removed using the Tweedie's formula for the Gaussian case and the Poisson noise is subsequentially reduced using the Tweedie's formula for the Poisson noise. As the same trained neural network is used for both steps, the computational complexity increase is negligible. The results in Table 4 indicated that our method still outperformed other methods.

Table 2: Comparison results in FMD data set with real noises in terms of PSNR (dB).

| Dataset | N2V | N2S | N2Same | Ours |
|---|---|---|---|---|
| Confocal MICE | 35.83 | 36.32 | 36.42 | **36.73** |

# 7 Case Study for Noise Statics and Noise Level Mismatch

In this section, we examined the mismatch of the noise statisticcs and the noise level between the training data and the inference level. For the mismatch of noise model, we carried out experiments in which the noise statistics of the training data differ from those in the inference phase. As shown in Table 5, all denoisers perform best when the noise statistics match in the training and inference phases. If the noise statistics differ between the two phases, we can observe performance deterioration in all methods. Nevertheless, the proposed method surpasses the other methods in almost the cases.

In the case of the mismatch of noise level, we additionally carried out experiments when there is a noise level mismatch between the training and inference phases. Table 6 shows comparison results using BSD68 dataset. Specifically, all methods are trained with a training data set corrupted with Poisson noise distribution with $\zeta = 0.01$. If the image at the inference phase is also corrupted with the Poisson noise with $\zeta = 0.01$, all methods show the best performance. However, as the noise level in the test data increases, their performance decreased. Nonetheless, our proposed method shows the most robust performance compared with other self-supervised methods.

Table 3: Comparison with different methods in the event of a discrepancy between the noise statistics of the training set and the test set in the CBSD68 data set.

| Test set of noisy type | Gaussian ($\sigma = 25$) | Poisson ($\zeta = 0.01$) | Gamma ($k = 100$) |
|---|---|---|---|
| Train set of noisy type | N2V / N2S / Ours | N2V / N2S / Ours | N2V / N2S / Ours |
| Gaussian ($\sigma = 25$) | 29.22 / 30.05 / 30.85 | 29.69 / 30.23 / 30.65 | 30.32 / 29.74 / 29.57 |
| Poisson ($\zeta = 0.01$) | 26.73 / 26.43 / 27.65 | 31.85 / 31.04 / 32.61 | 30.21 / 30.63 / 31.23 |
| Gamma ($k = 100$) | 25.74 / 26.51 / 27.95 | 29.06 / 30.34 / 29.34 | 31.14 / 30.54 / 33.82 |

Table 4: Quantitative results with various methods when there is a noise level mismatch between the training data set and test data in the BSD68 dataset.

| Method | Inference noise level $\zeta$ | | | | |
|---|---|---|---|---|---|
| | 0.01 | 0.02 | 0.03 | 0.04 | 0.05 |
| N2V | 28.73 | 26.67 | 24.79 | 23.39 | 22.32 |
| N2S | 29.76 | 27.03 | 24.73 | 23.01 | 21.71 |
| Ours | **30.81** | **29.01** | **28.00** | **27.25** | **26.64** |

# 8 Qualitative Results

We provided an additional qualitative comparison to valid the effectiveness of the proposed method. Fig. 2 illustrates the visual results of image denoising with various methods on the color image dataset, such as Kodak and CBSD68 dataset, when the noise parameters are known. Compared to the SOTA self-supervised learning approaches, our method provides 0.5∼3dB gain in PSNR with superb subjective image qualities.

Figs. 3, 4 shows the visual results in case of unknown noise parameter for gray-scale and color image dataset, respectively. Note that our blind approach provides comparable results to our method with known parameters. Moreover, the results are also comparable for supervised learning approach (supervised-blind) which is trained with various noise level inputs and clean references.

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

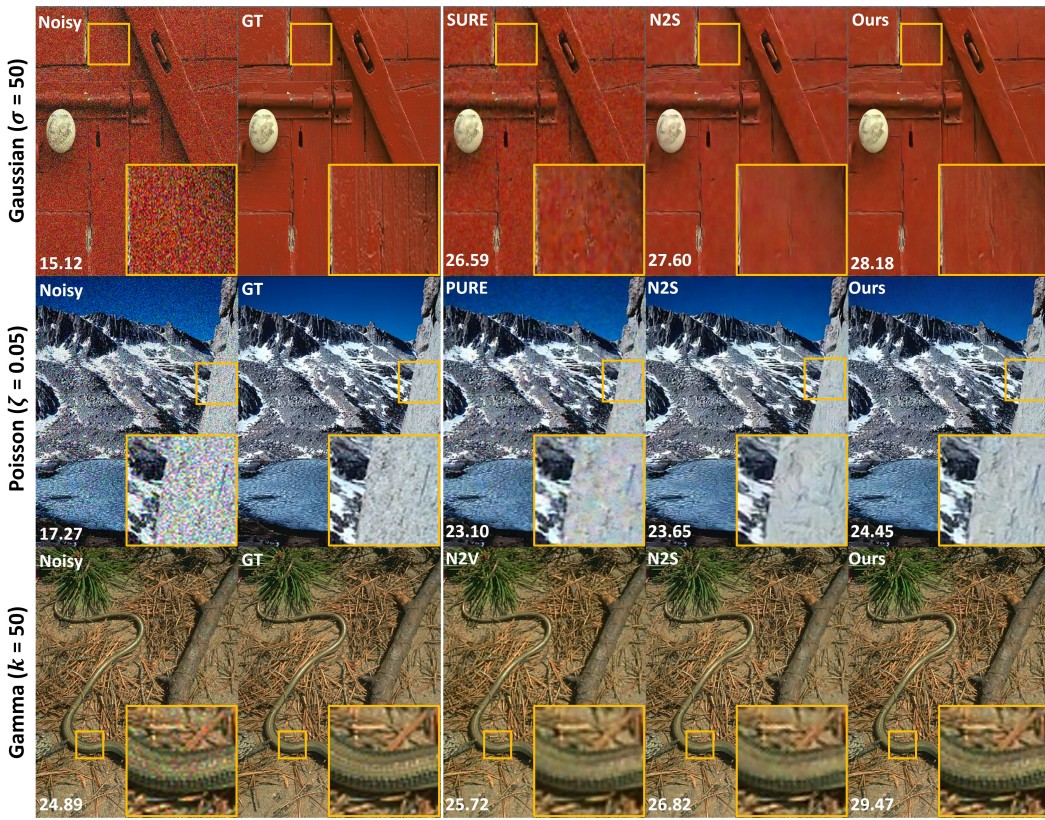

Figure 2: Qualitative comparison using Kodak and CBSD68 dataset. Top : Gaussian noise with $\sigma$= 50. Middle: Poisson noise with $\zeta = 0.05$. Bottom: Gamma noise with $k = 50$. White numbers at the lower left part of the images indicate the PSNR values in dB. Noisy: noisy input, SURE: Stein unbiased risk estimate based denoiser, PURE: Poisson SURE, N2V: Noise2Void, N2S: Noise2Self.

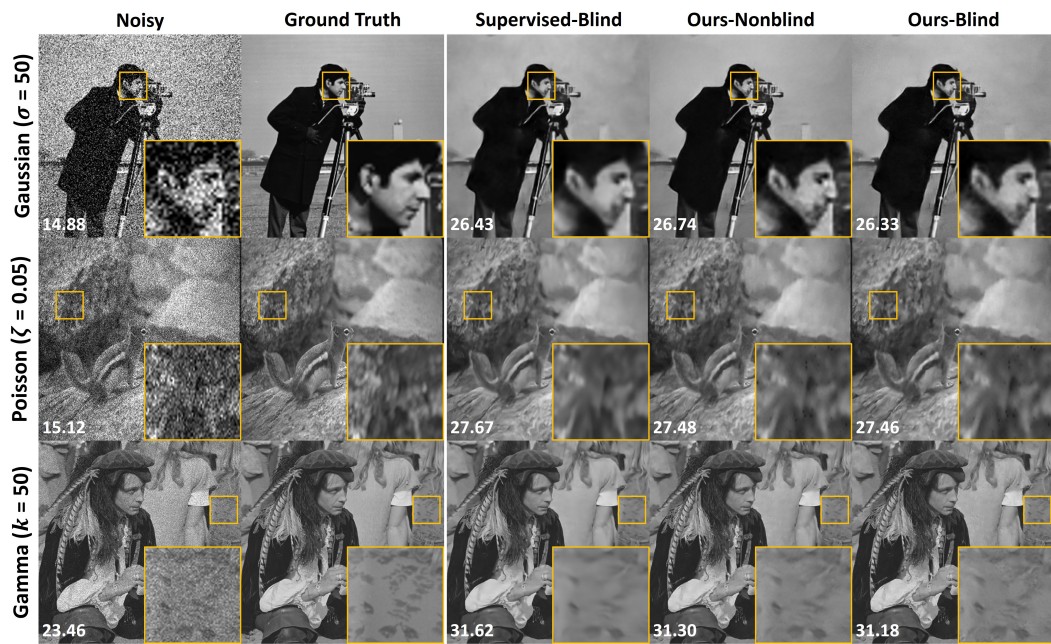

Figure 3: Qualitative comparison for gray-scale images using Set12 and BSD68 dataset when the noise parameters are unknown. Top : Gaussian noise with $\sigma$= 50. Middle: Poisson noise with $\zeta$ = 0.05. Bottom: Gamma noise with $k$ = 50. White numbers at the lower left part of the images indicate the PSNR values in dB. Note that our blind approach provides comparable results to our method with known parameters.

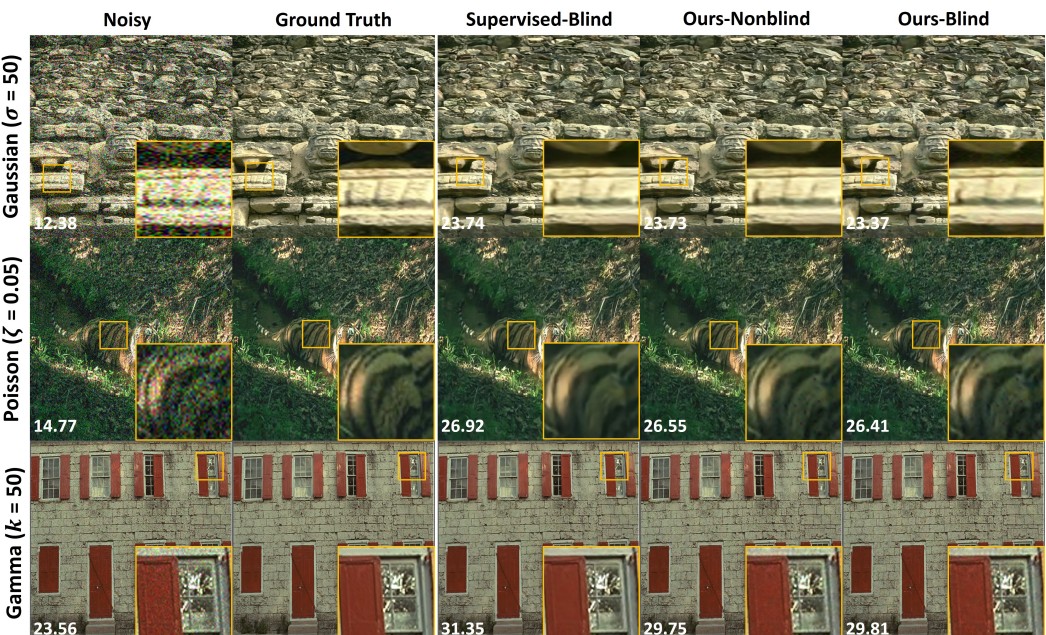

Figure 4: Qualitative comparison for color-scale using Kodak and CBSD68 dataset when the noise parameters are unknown. Top : Gaussian noise with $\sigma$= 50. Middle: Poisson noise with $\zeta$ = 0.05. Bottom: Gamma noise with $k$ = 50. White numbers at the lower left part of the images indicate the PSNR values in dB. Noisy: noisy input. Note that our blind approach provides comparable results to our method with known parameters.