# OpenReview forum: "Noise2Score: Tweedie’s Approach to Self-Supervised Image Denoising without Clean Images"
_NeurIPS.cc/2021/Conference — NeurIPS 2021 Poster_

### Official Review · Reviewer_1non · 2021-07-08

**Rating:** 7
**Confidence:** 4

**Summary:**

The authors how many recent self-supervised denoising methods can be related to Tweedie's formula and based on this insight they propose a novel method that can work with different noise distributions and outperform previous apporaches.

**Limitations And Societal Impact:**

Societal impact has been discussed.

**Main Review:**

Overall the paper is well written, even though some concepts can be hard to follow if the reader is not familiar with Tweedie's formula. The theoretical insight linking the formula to recent methods in the self-supervised denoising literature is interesting and novel and brings the authors to an original formulation of a denoising method. My main concern is the lack of discussion and comparisons with the work by Laine et al. "High-Quality Self-Supervised Deep Image Denoising", NeurIPS 2019, which showed how the various Noise2X methods are limited and also formulate a Bayesian framework leading to the estimation of the posterior mean of the image. While the two experimental settings are not directly comparable, it is seems that the work by Laine et al. already had performance competitive or superior to the one shown by the proposed method. The authors should address these points.

**Time Spent Reviewing:**

1

---

> ### Author Response · Authors · 2021-08-09
> **Response to Reviewer 1non**
>
> **$\#$ The lack of discussion and comparisons with the work by Laine et al. "High-Quality Self-Supervised Deep Image Denoising", NeurIPS 2019.**
>
> $\Rightarrow$ Per your suggestion, we carried out experimental comparisons with Laine et al. For a fair comparison, we trained all methods with the combined DIV2K and BSD400 dataset in gray scale. Then, the trained networks were used for inference using the BSD68 dataset for Gaussian noise and Poisson noise, as shown Table 1. The results confirmed that our method outperformed the two different implementations of Laine et al.
>
> Table 1. Comparison results in BSD68 dataset for the case of Gaussian noise and Poisson noise in terms of PSNR (dB).
>
> |    Noise type    | Laine19-mu | Laine19-pme |  Ours |
> |:----------------:|:----------:|:-----------:|:-----:|
> | Gaussian($\sigma$ = 25) |    28.13   |    29.04    | 29.12 |
> | Poisson($\zeta$ = 0.01) |    28.49   |    30.70    | 30.81 |

---

> > ### Comment · Reviewer_1non · 2021-08-27
> > **Response to authors**
> >
> > After reading all the authors' responses, I think they addressed the main points of concerns in a satisfactory manner. I raised my score accordingly.

---

> > > ### Author Response · Authors · 2021-08-28
> > > **Thanks for the positive feedback**
> > >
> > > We're happy to hear that our response adequately addressed the reviewer's concerns. Thank you for your understanding and the positive feedback on our updated experiments.

---

### Official Review · Reviewer_kPzN · 2021-07-16

**Rating:** 6
**Confidence:** 4

**Summary:**

This paper considers the problem of image denoising using deep learning. Traditionally, deep denoisers are trained in a supervised way using pairs of noisy and ground-truth images. Recent methods, such as Noise2Noise and those based on SURE, are used to train deep denoisers without ground truth.

This paper proposes Noise2Score as an alternative to existing methods. The idea of Noise2Score is to pre-train a denoising autoencoder (DAE) and use it as an estimator of the score function within the Tweedie’s formula. The key observation is that the pre-training stage of DAE doesn't explicitly need to know the true noise statistics. On other hand, the Tweedie’s formula enables the adaptation of the DAE to specific noise statistics.

The main conceptual contribution of the paper is that it establishes a formal link between the usage of DAEs for approximating score functions and image denoising. The empirical results show the competitive performance of Noise2Score against several well known image denoising methods, including Noise2Noise and SURE.

**Ethical Concerns:**

No issues with this.

**Limitations And Societal Impact:**

No issues with this.

**Main Review:**

Strengths: The key idea of Noise2Score—based on combining the DAE training with the Tweedie’s formula—is original in the context of denoising without ground truth. The method seems to perform well compared to related methods on simulations with several types of noise.

Major Issue 1: The current manuscript is vague on important theoretical/conceptual issues. (1) We know the theoretical conditions for both Noise2Noise and SURE to work, but not for Noise2Score. As an example, what are the assumptions necessary for the DAE to well approximate the score function? (2) How close should the noise statistics be to the one used in the Tweedie’s formula at the inference stage? If I believe the noise is AWGN and use Line 1 in Table 1, but the true noise is Poisson, how will this influence the result? (3) What is expected to happen if the training samples for p(y) in DAE training are not matched to the actual distribution of noise. (4) The noise estimation in (32) seems to use traditional denoising loss functions based on TV. Is there a reason one cannot leverage similar ideas used to develop Noise2Score? (5) The manuscript needs editing for clarity. The symbol sigma is overloaded as the noise variance and as a sampling parameter in (25). It is worth also mentioning that unlike Noise2Score, Noise2Noise does not need the knowledge of the statistical noise distribution at the inference stage.

Major Issue 2: Missing important details on empirical validation. (1) How was the UNet for Noise2Score trained. Was there only one Unet trained for all experiments or several? (2) How were the noisy images/patches sampled for training UNet? Were they picked equally from different distributions (Poisson, AWGN) or only from one? Was the training using only one level of noise or multiple? (3) Why wasn’t Noise2Noise tested on unknown parameters? Noise2Noise doesn’t need the knowledge of the parameter at the inference stage.

Summary: This paper is proposing an intriguing alternative to Noise2Noise and related methods. However, the paper falls short in terms of fully explaining the concept and its theoretical properties. Additionally, empirical validation overlooks important details on training, which makes it difficult to understand the potential of Noise2Score to actually work.


**Time Spent Reviewing:**

3 hours

---

> ### Author Response · Authors · 2021-08-09
> **Response to Reviewer kPzN**
>
> **$\#$ What are the theoretical conditions for Noise2Score? For example, what are the assumptions necessary for the DAE to well approximate the score function?**
>
> $\Rightarrow$ From [1,2], DAE provides an approximation of the score function whose approximation error is upper-bounded by o($\sigma^2$), where $\sigma$ is the additive noise level during training (See Eq. (23) in the paper). This implies that if $\sigma$ is sufficiently small, the approximation of the score function becomes more accurate. AR-DAE [2] further improves the accuracy of the score function approximation by making the estimate more stable for sufficiently small $\sigma$, which is why we used AR-DAE for the score function estimation. Since the accuracy of Tweedie’s formula depends on that of the score function estimate, we can bound the error from the exact Tweedie’s Bayesian estimate using the approximation error of the score function. Accordingly, in the final version, we will provide further discussion on the approximation error for each noise model.
>
> [1] Guillaume Alain et al. What regularized auto-encoders learn from the data-generating 314 distribution. The Journal of Machine Learning Research, 15(1):3563–3593, 2014.
>
> [2] Jae Hyun Lim, et.al. AR-DAE: Towards Unbiased 316 Neural Entropy Gradient Estimation. In International Conference on Machine Learning, pages 6061–6071. 317 PMLR, 2020.
>
> **$\#$ How close should the noise statics be to the one used in Tweedie’s formula at inference stage? If I believe the noise is AWGN and use Line 1 in Table 1, but the true noise is Poisson, how will this influence the result?**
>
> $\Rightarrow$ Thanks for your constructive comments. In order to respond to your comment, we carried out experiments in which the noise statistics of the training data differ from those in the inference phase. As shown in Table 1, all denoisers perform best when the noise statistics match in the training and inference phases. If the noise statistics differ between two phases, we can observe a deterioration in performance with all methods. Nevertheless, the proposed method surpasses the other methods in almost the cases.
>
> Table 1. Comparison with different methods in the event of a discrepancy between the noise statistics of the training set and the test set in the CBSD68 data set.
>
> |       Sampled as Input      |       Gaussian (σ =25)       |       Poisson (ζ= 0.01)      |         Gamma (k=100)        |
> |:---------------------------:|:----------------------------:|:----------------------------:|:----------------------------:|
> | Train dataset of noisy type |       N2V / N2S / Ours       |        N2V / N2S / Ours      |       N2V / N2S / Ours       |
> |       Gaussian (σ =25)      |     29.22 / 30.05 / 30.85    |     29.69 / 30.23 / 30.65    |     30.32 / 29.74 / 29.57    |
> |      Poisson (ζ = 0.01)     |     26.73 / 26.43 / 27.65    |     31.85 / 31.04 / 32.61    |     30.21 / 30.63 / 31.23    |
> |        Gamma (k=100)        |     25.74 / 26.51 / 27.95    |     29.06 / 30.34 / 29.34    |     31.14 / 30.54 / 33.82    |
>
>
> **$\#$ What is expected to happen if the training sampled for p(y) in DAE training are not matched to the actual distribution of noise?**
>
> $\Rightarrow$ Table 2 shows comparison results using BSD68 dataset when there is a noise level mismatch between the training and inference phases.  Specifically, all methods are trained with a training data set corrupted with Poisson noise distribution with $\zeta$ = 0.01. If the image at the inference phase is also corrupted with the Poisson noise with $\zeta$ = 0.01, all methods show the best performance. However, as the noise level in the test data increases, their performance decreased. Nonetheless, our proposed method shows the most robust performance compared with other self-supervised methods.
>
> Table 2. Quantitative results with various methods when there is a noise level mismatch between the training data set and test data in the BSD68 dataset.
>
> | Method | $\zeta$ = 0.01 | $\zeta$ = 0.02 | $\zeta$ = 0.03 | $\zeta$ = 0.04 | $\zeta$ = 0.05 |
> |:------:|:--------:|:--------:|:--------:|:--------:|:--------:|
> |   N2V  |   28.73  |   26.67  |   24.79  |   23.39  |   22.32  |
> |   N2S  |   29.76  |   27.03  |   24.73  |   23.01  |   21.71  |
> |  Ours  |   30.81  |   29.01  |   28.00  |   27.24  |   26.64  |
>
> **$\#$ The noise estimation in (32) seems to use traditional denoising loss functions based on TV. Is there a reason one cannot leverage similar ideas used to develop Noise2Score for estimating unknown parameter?**
>
> $\Rightarrow$ Thanks for the very insightful comments. One could also derive the Tweedie's formula in terms of the unknown noise parameter using the similar techniques in Section 3.1. A quick derivation shows that the resulting formula requires either knowledge of the unknown image or need two noisy measurements. That being said, there could be a walk-around that could be a very interesting research topic for future research.
>
> **$\#$ The manuscripts need editing for clarity for the symbol sigma and the property of Noise2Noise.**
>
> $\Rightarrow$ Thanks for the comment. We will use a different symbol for the sampling parameter sigma in (25) to avoid confusion due to the noise variance.  We will also explicitly mention that Noise2Noise does not require any prior information about the noise distribution.
>
> **$\#$ Was there only one U-Net trained for all experiments or several?**
>
> $\Rightarrow$ No. We trained a U-Net for every noise type. The description will be elaborated in the final version.
>
> **$\#$ How to sample the noisy image distribution in phase of training?**
>
> $\Rightarrow$ In the training phase, we only selected one noise distribution. For example, if we train Noise2Score for Gaussian noise, all training images are corrupted by Gaussian noise. In the case of the "known" parameter, during the training we sampled noisy images with only one noise level, but in the case of experiments with unknown noise parameters, we randomly sampled images with multiple noise levels during training, as reported in the main paper. This will be further clarified in the final version.
>
> **$\#$ Why wasn’t Noise2Noise tested on unknown parameters?**
>
> $\Rightarrow$ Although we had carried out experiments using Noise2Noise with unknown parameters, we did not include it to save spaces. Now Table 3 shows the comparison results including Noise2Noise with unknown noise level, which will be included in the final version.
>
> Table 3. Quantitative comparison with Noise2Noise in terms of PSNR (dB) when the noise parameters are unknown.
>
> |     Noise type    |        BSD68       |        Set12       |        CBSD68      |        Kodak       |
> |:-----------------:|:------------------:|:------------------:|:------------------:|:------------------:|
> |      Gaussian     |      Ours / N2N    |      Ours / N2N    |      Ours / N2N    |      Ours / N2N    |
> |       $\sigma$ = 25      |     28.95/28.98    |     30.08/30.08    |     30.78/30.91    |     31.78/31.96    |
> |       $\sigma$ = 50      |     25.81/28.86    |     26.65/26.72    |     27.32/27.66    |     28.83/29.07    |
> |       Poisson     |      Ours / N2N    |      Ours / N2N    |      Ours / N2N    |      Ours / N2N    |
> |      $\zeta$ = 0.01     |     30.36/30.91    |     31.42/31.67    |     32.23/32.95    |     32.96/33.85    |
> |      $\zeta$ = 0.05     |     26.77/27.19    |     27.76/28.01    |     28.23/29.07    |     28.98/30.13    |
> |        Gamma      |      Ours / N2N    |      Ours / N2N    |      Ours / N2N    |      Ours / N2N    |
> |       $k$ = 100     |     32.54/32.83    |     32.89/33.19    |     33.50/35.33    |     34.22/36.16    |
> |        $k$ = 50      |     30.42/31.07    |     30.83/31.58    |     30.93/33.41    |     31.42/34.39    |

---

> > ### Comment · Reviewer_kPzN · 2021-08-28
> > **Response to Rebuttal**
> >
> > Thank you for the detailed response! I am increasing my rating by one point to reflect the effort put by the authors on the rebuttal. However, this paper clearly needs a major revision, which is corroborated by the amount of information provided in the rebuttal, but missing in the original submission.

---

> > > ### Author Response · Authors · 2021-08-29
> > > **Thanks for increasing your rating!**
> > >
> > > Thanks for your positive feedback and for increasing your rating. We will ensure that all of the information contained in the rebuttal is incorporated into the final version of the paper.

---

### Official Review · Reviewer_yu4N · 2021-07-16

**Rating:** 6
**Confidence:** 5

**Summary:**

The paper proposes a new approach based on Tweedie's formula to perform self-supervised image denoising. The proposed approach has two steps: (1) learn the score function or gradient of density of the noisy images, and (2) use Tweedie's formula to compute a denoised image using the learned score function. The proposed approach builds on multiple relevant work in the literature, and is well motivated. The model is tested on Gaussian, Poisson and Gamma noise removal tasks on natural images and achieves competitive results.

**Limitations And Societal Impact:**

A major practical limitation, as the authors outlined is the assumption of a known noise type belonging to exponential distribution (and a way to estimate noise parameters at test time).

**Main Review:**

The paper is well organized, and clearly written. The experiments are well thought and supports the main claim of the paper. Please find my specific comments below:

1. **Missing baseline in Table 2**. The authors have done a thorough job of comparison, however, I think one of the relevant method is missing. N2V and N2S do not exploit noise type while learning the denoising function, and in general suffer from the drawback that they do not use the value of central pixel. Several post-processing schemes have been introduced to rectify this and incorporate the value of the central pixel back into the denoised output by assuming that the noise type is known. Given that the proposed method exploits the noise type, I think it is unfair to leave such methods out of comparison. For example, [High-Quality Self-Supervised Deep Image Denoising](https://arxiv.org/abs/1901.10277), Laine et. al, NeurIPS 2019 incorporates blind-spot in N2V and N2S into the architecture and has a post-processing scheme for Poisson and Gaussian noise. This post-processing scheme (again based on a Bayesian view) is shown to provide non-trivial improvement in denoising performance. When the noise type is unknown, Noise2Same [5] is a better baseline than N2V and N2S, since it promotes the use of center pixel.

2. **Relation with Noise2X**. I appreciate the authors providing context and illuminating the relationship of their method with existing methods in the literature. However, I think the relation with Noise2Void discussed in lines 191-198 is force fitting a narrative. The main idea behind masking schemes in Noise2Void and Noise2Self is to discourage the use of central pixel in the denoising function, and it looks like this is a non-trivial interpretation to get from score matching perspective. In my understanding, the practical version of masking used in Noise2Void, pixel values are replaced with a random pixel from the neighbourhood. However, the proposed method maybe more releted to Noiser2Noise (see below). Also, one of the other strong points claimed by the authors is the university of the score matching cost function for different noise types. But the relation is explained only using SURE. I think there is no easy interpretation for PURE or other noise types, which maybe worth highlighting in the paper for an informed reader.

3. **Noise2X and SURE beiing suboptimal for non-Gaussian**. In my understanding, Noise2Void or other blind-spot methods only make the assumption that the noise is pixel-wise independent. They are not even optimal for Gaussian denoising. SURE is derived explicitly assuming that the noise distribution is Gaussian, there are variants of SURE for most known noise types. See [Least squares estimation without priors or supervision. Raphan and Simoncelli. 2011](http://www.cns.nyu.edu/pub/eero/raphan10.pdf)

4. **Noiser2Noise**: I think the method is more related to [Noiser2Noise, Moran et. al, CVPR 2019](https://arxiv.org/pdf/1910.11908.pdf). The training process, and the estimation formula is very related. Readers may benefit if the authors include this in related work, and discuss more about it. It appears to me that Noiser2Noise essentially has the same idea derived without referring Tweedie's formula and score matching.

5. **Real data**. This is completely optional, but I would also appreciate if the authors tried to apply this method to real dataset. I understand that the method assumes known noise distributions, but some of the real datasets, like [fluorescence microscopy dataset](https://arxiv.org/abs/1812.10366), [electron-microscopy dataset](https://arxiv.org/pdf/2010.12970.pdf), or [DND and SIDD](https://arxiv.org/pdf/2003.07761.pdf) have real noise which can be modeled fairly accurately. Even if the authors choose not to include experiments on these datasets, they can discuss these datasets as potential real applications of their method.

**Minor nitpicking:**
Lines 221-234 makes it sounds like the proposed method makes it much easier to perform blind-denoising. However, for baselines like N2S or N2V or [Laine et. al](https://arxiv.org/abs/1901.10277) performing blind-denoising is easier than the proposed approach since they do not have to estimate the $\sigma$ parameter. For SURE, one can just train for the whole range of $\sigma$ (by using the ground-truth $\sigma$ value), and apply the trained network directly to the test image. Thus the baseline methods do not require a separate $\sigma$ estimation step. While I don't consider this as a major limitation, I think the authors should either tone done the claim, or acknowledge that competing methods can also perform blind-denoising just as/more easily.

Overall I enjoyed reading the paper. Thanks for the great work.


================== **Post-rebuttal** =================

Thank you for a detailed response to my comments. I'm happy with the responses, and I maintain my score.

**Time Spent Reviewing:**

6

---

> ### Author Response · Authors · 2021-08-09
> **Response to Reviewer yu4N**
>
> **$\#$ Missing baseline in Table 2. Comparison with “High-Quality Self-Supervised Deep Image Denoising” and “Noise2Same”.**
>
> $\Rightarrow$ Per your suggestion, we carried out experimental comparisons with Laine et al, and Noise2Same [5]. For a fair comparison, we trained all methods with the combined DIV2K and BSD400 dataset in gray scale. Then, the trained networks were used for inference using the BSD68 dataset for Gaussian noise and Poisson noise, as shown Table 1. The results confirmed that our method outperformed the existing ones including two different implementations of Laine et al.
>
> Table 1. Comparison results in BSD68 dataset for the case of Gaussian noise and Poisson noise in terms of PSNR (dB).
>
> |    Noise type   | Noise2Void | Noise2Self | Noise2Same | Laine19-mu |  Laine19-pme |   Ours |
> |:---------------:|:----------:|:----------:|:----------:|:----------:|:------------:|:------:|
> |  Gaussian($\sigma$ = 25) |    26.27   |    28.28   |    28.00   |    28.13   |     29.04    |  29.12 |
> | Poisson($\zeta$ = 0.01) |    28.73   |    29.73   |    29.32   |    28.49   |     30.70    |  30.81 |X
>
> **$\#$ Relation with Noise2X. The discussion in lines 191-198, which is discussed about the score matching with Noise2X. Furthermore, the relation is explained only using SURE.**
>
> $\Rightarrow$ Thanks your insightful comments. We agree with your point that the purpose of Noise2Void and Noise2Self masking schemes is not to use the center pixel, but to use randomly chosen neighborhood pixels. That being said, we think this step could also be viewed as injecting noise into the center pixel. However, the resulting noise at the center pixel is not additively Gaussian as in Noise2Score, since the location-dependent noises is not stationary. Therefore, we will remove this analogy in the final version. We also agree that Noiser2Noise is more closely related to our method, so we will discuss this relation in more detail in the final version.
> We will also note that the current interpretation of the loss function in score matching only applies to SURE for Gaussian noise, and for the case of PURE and other types of noise, our interpretation of their loss functions needs to be expanded.
>
> **$\#$ Noise2X and SURE being suboptimal for non-Gaussian. Noise2Void or other blind-spot methods only make the assumption that the noise is pixel-wise independent. They are not even optimal for Gaussian denoising. SURE is derived explicitly assuming that the noise distribution is Gaussian, there are variants of SURE for most known noise types.**
>
> $\Rightarrow$ As you pointed out, Noise2Void or other blind spot methods are not even optimal for Gaussian noise, so this will be highlighted in the final version to avoid possible confusion. In addition, we are aware that there are variants of SURE for other noise models, and the authors of [1] also argued that SURE is related to score matching. However, in contrast to SURE and other variants of SURE, whose cost function varies depending on the type of noise, our method uses the same cost function for neural network training, but only the post-processing step differs depending on the type of noise. This is what we meant by universal neural estimation, which is believed to be one of the main advantges.  We will highlight this difference in the final version.
>
> [1] Least squares estimation without priors or supervision. Raphan and Simoncelli. 2011
>
> **$\#$ Comparison with Noiser2Noise:**
>
> $\Rightarrow$ In terms of adding extra noise to a noisy image, we agree that our method is closely related to Noisier2Noise. However, in the training phase of Noisier2Noise, the main assumption is that noise from the same noise statistic as the original noisy image should be injected into the noisy images so that the neural networks learn to reduce the noise from the noisy input image. In our Noise2Score training method, however, Gaussian noise is added to noisy images at different noise levels regardless of the noise statistics to estimate the score function rather than noise. In addition, in contrast to Noisier2Noise, our method has a post-processing step with the Tweedie’s formula, which is determined by the noise model such as Poisson, Gamma, etc. Accordingly, as shown in Table 2 above, our proposed method outperforms the Noisier2Noise result.
>
> Table 2. Comparison results in BSD68 dataset for the case of Gaussian noise in terms of PSNR (dB).
>
> |   Noise type   | Noisier2Noise |  Ours |
> |:--------------:|:-------------:|:-----:|
> | Gaussian($\sigma$ = 25) |     28.01     | 29.12 |
>
> **$\#$ Application for Real data such as fluorescence microscopy dataset.**
>
> $\Rightarrow$ At your suggestion, we carried out the experiments with real fluorescence microscopy data sets (FMD). See Table 3.  Here, we have taken raw noisy images from the confocal FISH categories, which consisted of 1000 images. We used the 900 images as a training data set and the remaining 100 images as a test data set. In the FMD data set, the noisy images are modeled with mixed Poisson-Gaussian noise. Accordingly, we used a two-step approach in which the Gaussian noise is first removed using the Tweedie’s formula for the Gaussian case and the Poisson noise is subsequentially reduced using the Tweedie's formula for the Poisson noise. As the same trained neural network is used for both steps, the computational complexity increase is negligible. The results in Table 3 indicated that our method still outperformed other methods.
>
> Table 3. Comparison results in FMD data set with real noises in terms of PSNR (dB).
>
> |   Confocal MICE | Noise2Self | Noise2Void | Noise2Same |  Ours |
> |:---------------:|:----------:|:----------:|:----------:|:-----:|
> |       PSNR      |    36.32   |    35.83   |    36.42   | 36.73 |
>
> **$\#$ Minor nitpicking: “Lines 221-234 makes it sounds like the proposed method makes it much easier to perform blind-denoising. However, for baselines like N2S or N2V or Laine et. al performing blind-denoising is easier than the proposed approach since they do not have to estimate the σ parameter”**
>
> $\Rightarrow$ Thanks for the constructive comments. Similar to Laine et al. we could also train the neural network to estimate the σ-parameter, so that this blind denoising becomes much easier. That being said, this is a future research topic, and in the final version we will tone down the claim and acknowledge that competing methods can also do blind denoising as/more easily.

---

> ### Author Response · Authors · 2021-08-30
> **We are looking forward to your feedback!**
>
> Dear Reviewer,
>
> Thanks again for your constructive comments. We kindly remind you that we have done our best to address your concerns with additional experiments, etc.  Since we only have about 3 days for discussions, your timely feedback would be highly appreciated. Could you please go over our responses and let us know if there are any remaining issues that we need to address?
>
> Best regards, Authors

---

### Author Response · Authors · 2021-08-27
**The end of discussion phase is approaching**

Dear Reviewers,

Thanks again for your constructive reviews. Could you please go over our responses as the discussion period is only one week left? We have tried our best to faithfully respond to your comments and did our best to provide additional experimental results per your suggestions. We sincerely appreciate your time and efforts in reviewing our paper and am looking forward to your response.

Best regards,
Authors

---

### Decision · Program_Chairs · 2021-09-27

**Decision:**

Accept (Poster)

**Comment:**

This paper proposes a new method, Noise2Score, for denoising images without clean references at training time. The proposed approach has two steps. First a denoising autoencoder (DAE) is used to learn the score function in a noise agnostic fashion. Second, the denoised image is computed through the estimated score function by exploiting Tweedie’s formula (which can be used with any exponential family noises). Empirical evaluation shows the benefits of the proposed approach (comparing to a large number of relevant baselines)

All reviewers found the proposed method interesting and novel. Furthermore, its connection with the recent literature in the subject (e.g. Noise2X) is a nice feature that will inform future research on the topic.

In the original review, several clarifications were needed. The author provided a detailed response which clarified many of the raised concerns. After the rebuttal, all reviewers recommend accepting the paper. Reviewers kPzN and 1non updated their score (to 6 and 7 respectively).

In sum, the work provides an interesting contribution and the current empirical evaluation is sufficient for accepting the paper. The AC encourages the authors to incorporate the detailed clarifications provided to the reviewers.